# Assessing the use of prescription drugs and dietary supplements in obese respondents in the National Health and Nutrition Examination Survey

**Laura A. Barrett**[1], **Aiwen Xing**[2], **Julia Sheffler**[3], **Elizabeth Steidley**[1], **Terrence J. Adam**[4], **Rui Zhang**[4], **Zhe He**[1,3]*

**1** School of Information, Florida State University, Tallahassee, Florida, United States of America,
**2** Department of Statistics, Florida State University, Tallahassee, Florida, United States of America,
**3** Department of Behavioral Sciences and Social Medicine, College of Medicine, Florida State University, Tallahassee, Florida, United States of America, **4** Institute for Health Informatics and College of Pharmacy, University of Minnesota, Minneapolis, Minnesota, United States of America

* zhe@fsu.edu

## Abstract

### Introduction

Obesity is a common disease and a known risk factor for many other conditions such as hypertension, type 2 diabetes, and cancer. Treatment options for obesity include lifestyle changes, pharmacotherapy, and surgical interventions such as bariatric surgery. In this study, we examine the use of prescription drugs and dietary supplements by the individuals with obesity.

### Methods

We conducted a cross-sectional analysis of the National Health and Nutrition Examination Survey (NHANES) data 2003–2018. We used multivariate logistic regression to analyze the correlations of demographics and obesity status with the use of prescription drugs and dietary supplement use. We also built machine learning models to classify prescription drug and dietary supplement use using demographic data and obesity status.

### Results

Individuals with obesity are more likely to take cardiovascular agents (OR = 2.095, 95% CI 1.989–2.207) and metabolic agents (OR = 1.658, 95% CI 1.573–1.748) than individuals without obesity. Gender, age, race, poverty income ratio, and insurance status are significantly correlated with dietary supplement use. The best performing model for classifying prescription drug use had the accuracy of 74.3% and the AUROC of 0.82. The best performing model for classifying dietary supplement use had the accuracy of 65.3% and the AUROC of 0.71.

**Data Availability Statement:** All the data files are available from the National Health and Nutrition

Examination Survey (NHANES) database at https://www.cdc.gov/nchs/nhanes/index.htm.

**Funding:** This study was supported in part by the National Institute on Aging (NIA) of the National Institutes of Health (NIH) under Award Number R21AG061431 (ZH); and the University of Florida Clinical and Translational Science Institute, which is supported in part by the NIH National Center for Advancing Translational Sciences under award number UL1TR001427 (ZH); and National Center for Complementary and Integrative Medicine (NCCIH) of NIH under Award Number R01AT009457 (RZ). The content is solely the responsibility of the authors and does not necessarily represent the official views of the NIH. The funders had no role in study design, data collection and analysis, decision to publish, or preparation of the manuscript.

**Competing interests:** The authors have declared that no competing interests exist.

## Conclusions

This study can inform clinical practice and patient education of the use of prescription drugs and dietary supplements and their correlation with obesity.

## Introduction

As a major health and economic crisis affecting the modern world, much progress has been made in identifying and developing strategies for preventing and treating obesity. Currently, treatment options include lifestyle changes, pharmacotherapy, and surgical interventions (e.g., intragastric balloons and bariatric surgery) [1]. In terms of pharmacotherapy, there are five approved prescription drugs (RXD) (orlistat, 1999; phentermine/topiramate, 2012; liraglutide, 2014; naltrexone/bupropion, 2014; and semaglutide, 2021) that can be prescribed for weight loss [2]. All but orlistat, which reduces the absorption of fat, work by helping the individual to limit caloric intake [3]. There are also four RXD that are similar to amphetamines that can be used short-term (phendimetrazine, diethylpropion, phentermine, and benzphetamine) [4]. There have been three other well-known RXD that were approved for use and then removed from the market. The first one is fenfluramine/phentermine (fen-phen) which was discontinued in 1997 because fenfluramine was shown to cause cardiac issues [5]. The second one is sibutramine, which was withdrawn in 2010 due to an increased risk of stroke and myocardial infarction [6]. The third one is lorcaserin, which was withdrawn in February 2020 after a clinical trial showed an increased occurrence of cancers [7]. Due to the cost of pharmacotherapy and surgical interventions, as well as other reasons, dietary supplements (DS) are often used as a cost-sensitive and easily accessible, albeit less scientifically supported, alternative treatment of obesity [8,9].

Individuals with obesity face an increased risk of chronic diseases, namely depression, type 2 diabetes, cardiovascular disease, and many cancers including those of the colon, breast, kidney, and pancreas [10]. Many of these conditions require pharmaceutical intervention as part of the treatment plan and individuals with obesity often use RXD to manage these conditions. Overall, RXD use in the United States has increased [9]. This increase is partly influenced by the development of new RXD, the expansion of RXD coverage by insurance companies, and increased rates of chronic conditions such as obesity [9]. The greatest increase in RXD use has been in those used for treating conditions found to be associated with obesity, specifically antihypertensives, antihyperlipidemic, antidiabetics, and antidepressants [11].

Recent studies have researched obesity in relation to specific drugs or drug types [12–16]. There have also been recent studies that examined various aspects of obesity such as childhood obesity [16,17], obesity and hours spent at work [18], exposure to certain pollutants or chemicals [19–21], trends in obesity [22], and obesity and waist circumference [23]. However, there have not been any studies that look at overall RXD use in individuals with obesity. Being able to see this bird's eye view of this relationship is important because understanding the patterns of RXD use among people with obesity, who often have other chronic conditions, can inform both clinical practice and research [9]. This is challenging because there are cross-over issues between RXD, their side effect of weight gain, and their therapeutic effect on obesity and its comorbidities. For example, certain blood-glucose-lowering RXD and psychotropics may lead to unintended weight gain [24]. In this project, we aim to gain an in-depth understanding of both the relationship between obesity and RXD use, as well as the correlations between specific RXD and DS use in individuals with obesity. We also aim to understand if demographic

variables and obesity status can assist with classifying an individual's likelihood of using any RXD or DS, not just specific RXD or DS.

## Materials and methods

The National Center for Health Statistics of CDC has been conducting the National Health and Nutrition Examination Survey (NHANES) as a continuous cross-sectional health survey [25]. It samples the non-institutionalized population of the United States with a stratified multistage probability model and releases results of a set of health surveys, medical examinations, a physical, and laboratory test every two years. Its rigorous quality control ensures high-quality data collection and national representativeness. The NHANES data have been used in many public health and epidemiology studies [26–32].

Demographic, physical examination, prescription drug (RXD) and dietary supplement (DS) use [8], and health insurance information were extracted from NHANES for survey years 2003–2018 (8 survey cycles). The 16-year sample weight (the number of people in the US population that a sample in the combined sample can represent) was calculated according to the analytical guideline of NHANES [25]. The obese group is defined as: (1) BMI $\geq$ 30 kg/m$^2$ [33], (2) age $\geq$ 18 [34]. From the original NHANES data, 2689 respondents with no BMI and 45 respondents with no RXD use information were removed from the dataset. S1 Table in the supplementary material lists the NHANES file, the NHANES variable names, the associated questions, and how they are referred to in this paper.

### Data analysis

**Basic characteristics.** A profile for each group was created that included sex, age, race, annual household income, and health insurance status.

**Statistical analysis.** We conducted multivariate logistic regression analyses to access: 1) the associations between using RXD/DS and variables of interests (i.e., demographic characteristics, poverty income ratio, insurance status, and obesity status) and 2) the associations between taking specific types of RXD and obesity status. Weighted multivariate logistic regression analyses were used to obtain odds ratios (OR) and 95% CIs with 16-year sample weight. All interested variables were introduced in the model first then backward elimination with a threshold of p = 0.05 was applied to eliminate variables. We kept only the variables that were significant in the initial model in the final model. The significance level was set as 0.05. All statistical analyses were performed by SAS software (SAS Institute Inc), version 9.4.

We performed two separate logistic regression analyses. 1) Usage based on the specific number of RXD/DS that the individual used was the dependent variable. Demographic characteristics were included as independent variables to examine whether taking a specific number of RXD/DS was significantly associated with demographic characteristics within non-obese or obese groups separately. This regression evaluated covariates down to the two groups. 2) Obesity status and demographic characteristics were included as independent variables to test their associations with taking a specific number of RXD/DS within the whole population. In the second analysis, the dichotomous dependent variable was whether participants were prescribed the specific types of RXD; obesity status was set as independent variable with reference as non-obese group.

**Classification modeling.** We used Weka [35] to evaluate different machine learning models for classifying whether a respondent used one or more RXD or DS, respectively. In the first round of the modelling, the variables included age group, sex, BMI category, race, annual household income, and insurance status. As we are also interested in seeing whether DS use would help classify RDX use and vice versa, in the second round of modeling, DS use or RXD

use was added as a variable for classifying the use of the other type. A third round of modelling was done using the poverty income ratio (PIR: a ratio of family income to poverty threshold) in place of the annual household income. Lastly, we further evaluated if machine learning models were able to classify how many RXD were used. For this round, we created four groups of RDX count (i.e., 0, 1–2, 3–5, >5). We used feature selection based on correlation ("CorrelationAttributeEval" in Weka) to rank the importance of the variables. We evaluated four major machine learning algorithms including Naïve Bayes, Logistic Regression, SMO (Weka's implementation of Support Vector Machine), and Random Forest. Deep learning techniques were not employed because of the small number of variables in this dataset. The data was preprocessed to make all numerical data nominal. 10-fold cross validation was employed. In each fold, 90% of the data was used for training and 10% of the data was used for testing. The models were compared using overall accuracy, precision, recall, F1-score, and AUROC.

## Results

### Basic characteristics

Table 1 shows the basic characteristics of the two groups regarding their RXD use. There are a few differences based on obesity status and demographics. In the non-obese group 52.59% of people report taking 1 or more RXD. The obese group has a higher reported use at 63.84%. In both groups, females report higher use than males. In addition, RXD use increases with age in both groups. Race also plays a role in reported RXD use in both groups; Non-Hispanic Whites have the highest percentages of use while Mexican Americans show the lowest percentages. Lastly, in both groups, those with health insurance reported higher RXD use than those that reported not having insurance.

### Specific RXD types

For the identified RXD types, we evaluated the association between the specific types of RXD use and obesity status. Table 2 shows the odds ratio between obese and non-obese group taking prescription drugs. While looking at the types of RXD, there were differences in use between the obese and non-obese groups. Cardiovascular agents are the most used RXD type in both groups. This is not surprising given the prevalence of cardiovascular diseases in the United States. Compared with those in the non-obese group, individuals with obesity are more likely to take cardiovascular agents (OR = 2.095, 95% CI 1.989–2.207) and metabolic agents (OR = 1.658, 95% CI 1.573–1.748). Fig 1(a) shows a comparison of the cardiovascular agents based on percent of the RXD used stratified by each age group and weight category (underweight, normal weight, overweight, and obese. Fig 1(b) shows a comparison of the metabolic agents used stratified by age group and weight category. In both types of RXD, the obese population uses more of the RXD in the younger age groups, starting clearly at 25–34. The decrease in usage by obese individuals in the 75+ group may, in part, reflect the potential mixed effects of obesity in old age. Studies indicate that people with higher than normal BMI have lower mortality in the 75+ age group, though this is dependent on many other variables [36–38].

### Correlation analysis of RDX and DS Use with demographic characteristics

Examining the correlation between reported RDX use and demographic characteristics for the non-obese and obese groups, we found a few items of interest. S2 Table shows the results of this regression analysis. Male from both the non-obese and the obese group were significantly less likely than female to use RXD ($OR_{control}$ = 0.53, 95% CI 0.498–0.564, $OR_{obese}$ = 0.569, 95% CI 0.52–0.623). Compared with those individuals with obesity older than 75, adults younger

**Table 1. Basic characteristics of the study population.**

| Variable | Non-obese | | | | Obese | | | |
|---|---|---|---|---|---|---|---|---|
| | 0 RXD | | 1 or more RXD | | 0 RXD | | 1 or more RXD | |
| | Weight Count | % | Weight Count | % | Weight Count | % | Weight Count | % |
| Total | 66033363 | 47.41% | 73250493 | 52.59% | 28449346 | 36.16% | 50222655 | 63.84% |
| Total Count | 14092 | 49.27% | 14508 | 50.73% | 6119 | 37.25% | 10310 | 62.75% |
| **Gender** | | | | | | | | |
| Male | 37570793 | 54.85% | 30921089 | 45.15% | 15263957 | 41.90% | 21161327 | 58.10% |
| Male Count | 7941 | 54.27% | 6692 | 45.73% | 3005 | 41.85% | 4175 | 58.15% |
| Female | 28462570 | 40.21% | 42329405 | 59.79% | 13185389 | 31.21% | 29061328 | 68.79% |
| Female Count | 6151 | 44.04% | 7816 | 55.96% | 3114 | 33.67% | 6135 | 66.33% |
| **Age** | | | | | | | | |
| 18–24 | 14166880 | 69.26% | 6287886 | 30.74% | 4828265 | 70.30% | 2040083 | 29.70% |
| 18–24 Count | 3708 | 75.94% | 1175 | 24.06% | 1214 | 73.53% | 437 | 26.47% |
| 25–34 | 17173005 | 66.88% | 8505063 | 33.12% | 8170177 | 63.04% | 4789193 | 36.96% |
| 25–34 Count | 3383 | 71.93% | 1320 | 28.07% | 1680 | 66.77% | 836 | 33.23% |
| 35–44 | 14370023 | 59.67% | 9713144 | 40.33% | 7134034 | 46.66% | 8154368 | 53.34% |
| 35–44 Count | 2784 | 65.11% | 1492 | 34.89% | 1418 | 50.00% | 1418 | 50.00% |
| 45–54 | 11133113 | 44.23% | 14038322 | 55.77% | 4884217 | 30.47% | 11146006 | 69.53% |
| 45–54 Count | 2009 | 48.96% | 2094 | 51.04% | 958 | 33.22% | 1926 | 66.78% |
| 55–64 | 6198064 | 31.45% | 13511728 | 68.55% | 2505342 | 17.60% | 11727619 | 82.40% |
| 55–64 Count | 1316 | 33.16% | 2653 | 66.84% | 583 | 19.56% | 2398 | 80.44% |
| 65–74 | 2030697 | 15.24% | 11290366 | 84.76% | 714089 | 8.08% | 8126671 | 91.92% |
| 65–74 Count | 565 | 17.04% | 2750 | 82.96% | 195 | 8.71% | 2043 | 91.29% |
| 75 and over | 961580 | 8.85% | 9903984 | 91.15% | 213221 | 4.79% | 4238715 | 95.21% |
| 75 and over Count | 327 | 9.76% | 3024 | 90.24% | 71 | 5.37% | 1252 | 94.63% |
| **Race** | | | | | | | | |
| Mexican American | 7643611 | 70.58% | 3186177 | 29.42% | 4773068 | 60.80% | 3077063 | 39.20% |
| Mexican American Count | 2879 | 64.39% | 1592 | 35.61% | 1616 | 52.95% | 1436 | 47.05% |
| Other Hispanic | 4645998 | 62.49% | 2788237 | 37.51% | 2239910 | 52.63% | 2016083 | 47.37% |
| Other Hispanic Count | 1366 | 54.90% | 1122 | 45.10% | 643 | 43.45% | 837 | 56.55% |
| Non-Hispanic White | 38690156 | 40.72% | 56324147 | 59.28% | 15007808 | 29.36% | 36107896 | 70.64% |
| Non-Hispanic White Count | 4696 | 37.94% | 7680 | 62.06% | 1758 | 27.01% | 4750 | 72.99% |
| Non-Hispanic Black | 7827436 | 58.04% | 5659904 | 41.96% | 4831848 | 41.64% | 6772380 | 58.36% |
| Non-Hispanic Black Count | 2953 | 54.23% | 2492 | 45.77% | 1695 | 38.15% | 2748 | 61.85% |
| Other Race or Multi-Racial | 7226162 | 57.73% | 5292029 | 42.27% | 1596713 | 41.52% | 2249232 | 58.48% |
| Other Race or Multi-Racial Count | 2198 | 57.54% | 1622 | 42.46% | 407 | 43.02% | 539 | 56.98% |
| **Household Income** | | | | | | | | |
| $0 to $34,999 | 18871838 | 47.76% | 20642494 | 52.24% | 8786323 | 36.67% | 15177268 | 63.33% |
| $0 to $34,999 Count | 5322 | 48.16% | 5728 | 51.84% | 2460 | 36.45% | 4289 | 63.55% |
| $35,000 to $74,999 | 19147850 | 48.23% | 20556912 | 51.77% | 9374135 | 37.16% | 15854727 | 62.84% |
| 35,000 to $74,999 Count | 3850 | 50.25% | 3812 | 49.75% | 1849 | 39.01% | 2891 | 60.99% |
| $75,000 and over | 21861695 | 45.06% | 26655556 | 54.94% | 7853627 | 33.23% | 15782826 | 66.77% |
| $75,000 and over Count | 3352 | 47.98% | 3634 | 52.02% | 1193 | 35.04% | 2212 | 64.96% |
| **Poverty Income Ratio** | | | | | | | | |
| 0-.99 | 11985065 | 54.42% | 10038503 | 45.58% | 5239978 | 40.20% | 7794072 | 59.80% |
| 0-.99 Count | 3443 | 55.31% | 2782 | 44.69% | 1520 | 40.90% | 2196 | 59.10% |
| 1–1.99 | 14166723 | 50.07% | 14127988 | 49.93% | 6774522 | 38.90% | 10642021 | 61.10% |

*(Continued)*

**Table 1.** (Continued)

| Variable | Non-obese | | | | Obese | | | |
|---|---|---|---|---|---|---|---|---|
| | 0 RXD | | 1 or more RXD | | 0 RXD | | 1 or more RXD | |
| | Weight Count | % | Weight Count | % | Weight Count | % | Weight Count | % |
| 1–1.99 Count | 3471 | 49.70% | 3513 | 50.30% | 1630 | 38.87% | 2563 | 61.13% |
| 2–2.99 | 9141944 | 46.30% | 10604451 | 53.70% | 4248535 | 35.37% | 7763139 | 64.63% |
| 2–2.99 Count | 1810 | 46.98% | 2043 | 53.02% | 820 | 35.56% | 1486 | 64.44% |
| 3–3.99 | 7371681 | 47.32% | 8205982 | 52.68% | 3044609 | 33.21% | 6121952 | 66.79% |
| 3–3.99 Count | 1233 | 47.31% | 1373 | 52.69% | 529 | 34.33% | 1012 | 65.67% |
| 4–4.99 | 5149047 | 43.13% | 6790653 | 56.87% | 2275579 | 31.94% | 4847852 | 68.06% |
| 4–4.99 Count | 821 | 44.94% | 1006 | 55.06% | 335 | 32.06% | 710 | 67.94% |
| 5 | 10758790 | 39.64% | 16379571 | 60.36% | 3821273 | 31.85% | 8178192 | 68.15% |
| 5 Count | 1558 | 41.74% | 2175 | 58.26% | 537 | 31.98% | 1142 | 68.02% |
| **Health Insurance**[*] | | | | | | | | |
| Yes | 46794083 | 41.29% | 66549204 | 58.71% | 19812159 | 30.28% | 45614641 | 69.72% |
| Yes Count | 9206 | 41.52% | 12968 | 58.48% | 3912 | 29.92% | 9163 | 70.08% |
| No | 19073757 | 74.28% | 6604505 | 25.72% | 8582725 | 65.52% | 4516163 | 34.48% |
| No Count | 4830 | 76.10% | 1517 | 23.90% | 2194 | 66.04% | 1128 | 33.96% |

[*]Overall insurance coverage was 82.0% covered and 17.8% not covered.

than 54 were significantly less likely to use RXD. When controlling for all other variables, non-obese people covered by insurance were around 1.597 times (p < .0001) as likely to use RXD than those who did not have any insurance coverage. Specifically, individuals with obesity covered by Medicare were 3.657 times more likely to use RXD (p < .0001) than those with no Medicare covered.

When looking at the correlation between reported DS use and demographic characteristics for the non-obese and obese groups, we found a few interesting findings (S3 Table). Within both groups, Mexican American and Non-Hispanic Black were significantly less likely to take DS compared with Non-Hispanic White. Individuals with obesity covered by private insurance, Medicare, and other government insurance were significantly more likely to take DS, while individuals without obesity covered by insurance were 1.411 times as likely to take DS than those who were not covered by any insurance while holding other variables constant.

**Table 2. Odds ratios between obese and non-obese group taking prescription drugs.**

| Prescription drugs | Obese (%) | Non-Obese (%) | Odds ratios | 95% Wald CI | | Pr > ChiSq |
|---|---|---|---|---|---|---|
| Cardiovascular agents | 37.11 | 21.01 | 2.095 | 1.989 | 2.207 | < .0001 |
| Metabolic agents | 28.02 | 16.86 | 1.658 | 1.573 | 1.748 | < .0001 |
| Central nervous system agents | 23.91 | 17.61 | 1.187 | 1.126 | 1.252 | < .0001 |
| Gastrointestinal agents | 14.21 | 9.62 | 1.277 | 1.199 | 1.36 | < .0001 |
| Respiratory agent | 9.46 | 6.59 | 1.214 | 1.128 | 1.306 | < .0001 |
| Psychotherapeutic agents | 15.16 | 10.13 | 1.304 | 1.226 | 1.387 | < .0001 |
| Hormones/hormone modifiers | 15.16 | 13.76 | 0.877 | 0.827 | 0.93 | < .0001 |
| Topical agents | 5.77 | 4.7 | 1.011 | 0.925 | 1.105 | 0.8118 |
| Anti-infectives | 5.5 | 5.56 | 0.795 | 0.729 | 0.868 | < .0001 |
| Coagulation modifiers | 4.41 | 2.69 | 1.372 | 1.233 | 1.526 | < .0001 |
| Antineoplastics | 1.48 | 1.46 | 0.83 | 0.705 | 0.977 | 0.0248 |

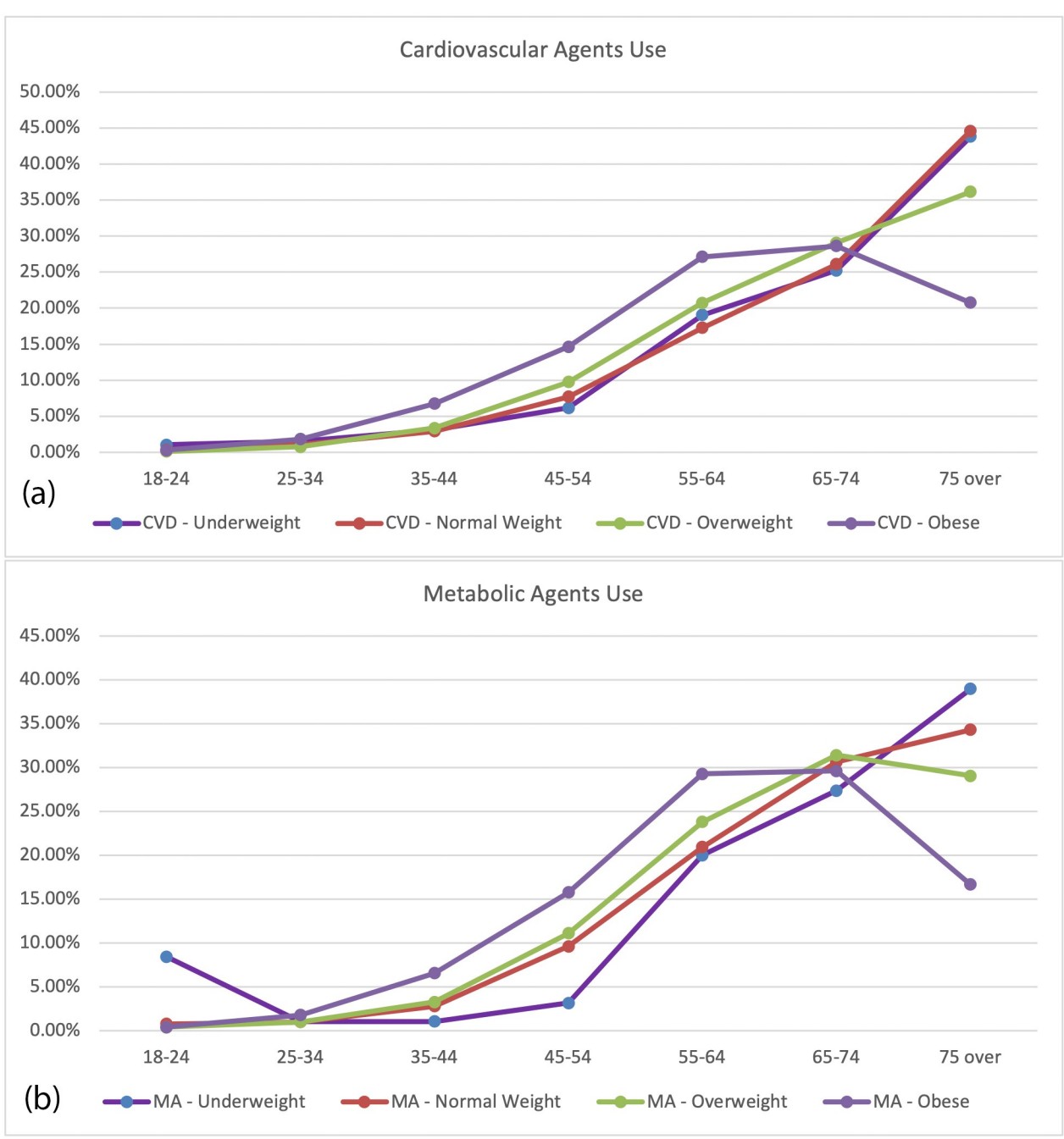

**Fig 1. Comparison of RXD used by age group and weight category.** CVD is cardiovascular agents. MA is metabolic agents. (a) Comparison of CVD RXD used by different age groups and weight categories. (b) Comparison of MA RXD used by different age groups and weight categories.

We were also interested in how obesity status and demographic characteristics associated with the use RXD or that of DS within the whole population group (Table 3). Female and older people were more likely to take RXD or DS. PIR is an interesting factor, as people with higher PIR were significantly more likely to take DS. For RXD use, only those with PIR from 1 to 2 were found to be statistically significant. This was also the case with the PIR showing that those with a PIR higher than 1 are more likely to take DS. The higher the PIR is, the higher

**Table 3. Reported prescription drug use/dietary supplements use by demographic characteristics and obesity status correlates.**

| Variable | RXD use | | | DS use | | |
|---|---|---|---|---|---|---|
| | Odds ratio | 95% Wald CI | P value | Odds ratio | 95% Wald CI | P value |
| **Gender** | | | | | | |
| Male | 0.543 | 0.516 0.572 | < .0001 | 0.574 | 0.548 0.602 | < .0001 |
| Female (reference) | 1 | | | 1 | | |
| **Age Group** | | | | | | |
| 18–24 | 0.1 | 0.081 0.124 | < .0001 | 0.256 | 0.219 0.301 | < .0001 |
| 25–34 | 0.127 | 0.103 0.157 | < .0001 | 0.349 | 0.299 0.408 | < .0001 |
| 35–44 | 0.189 | 0.153 0.233 | < .0001 | 0.397 | 0.341 0.463 | < .0001 |
| 45–54 | 0.325 | 0.263 0.401 | < .0001 | 0.534 | 0.459 0.621 | < .0001 |
| 55–64 | 0.592 | 0.479 0.733 | < .0001 | 0.802 | 0.689 0.933 | 0.0043 |
| 65–74 | 0.742 | 0.606 0.908 | 0.0039 | 0.885 | 0.779 1.006 | 0.0614 |
| 75 over (reference) | 1 | | | 1 | | |
| **Race** | | | | | | |
| Mexican American | 0.442 | 0.4 0.487 | < .0001 | 0.586 | 0.535 0.642 | < .0001 |
| Other Hispanic | 0.536 | 0.476 0.605 | < .0001 | 0.757 | 0.677 0.845 | < .0001 |
| Non-Hispanic White (reference) | 1 | | | 1 | | |
| Non-Hispanic Black | 0.61 | 0.562 0.663 | < .0001 | 0.589 | 0.545 0.636 | < .0001 |
| Other Race—Including Multi-Racial | 0.542 | 0.489 0.6 | < .0001 | 0.902 | 0.82 0.993 | 0.0345 |
| **Poverty Income Ratio** | | | | | | |
| 0–1 (reference) | 1 | | | 1 | | |
| 1–2 | 0.913 | 0.842 0.989 | 0.0255 | 1.102 | 1.024 1.185 | 0.0097 |
| 2–3 | 0.972 | 0.888 1.064 | 0.5362 | 1.171 | 1.079 1.272 | 0.0002 |
| 3–4 | 0.914 | 0.83 1.006 | 0.066 | 1.325 | 1.213 1.448 | < .0001 |
| 4–5 | 0.984 | 0.885 1.094 | 0.7621 | 1.511 | 1.371 1.666 | < .0001 |
| > = 5 | 1.058 | 0.968 1.157 | 0.2155 | 1.758 | 1.618 1.91 | < .0001 |
| **Covered by Any Insurance** | | | | | | |
| Yes | 1.58 | 1.338 1.866 | < .0001 | 1.21 | 1.069 1.369 | 0.0025 |
| No (reference) | 1 | | | 1 | | |
| **Covered by Private Insurance** | | | | | | |
| Yes | 1.38 | 1.181 1.612 | < .0001 | 1.299 | 1.163 1.451 | < .0001 |
| No (reference) | 1 | | | 1 | | |
| **Covered by Medicare** | | | | | | |
| Yes | 2.779 | 2.346 3.292 | < .0001 | 1.245 | 1.097 1.413 | 0.0007 |
| No (reference) | 1 | | | 1 | | |
| **Covered by Medicaid** | | | | | | |
| Yes | 2.024 | 1.692 2.422 | < .0001 | 0.819 | 0.714 0.94 | 0.0045 |
| No (reference) | 1 | | | 1 | | |
| **Covered by Other Government Insurance** | | | | | | |
| Yes | 1.935 | 1.634 2.291 | < .0001 | 1.17 | 1.031 1.327 | 0.0149 |
| No (reference) | 1 | | | 1 | | |
| **Obesity** | | | | | | |
| Yes | 1.565 | 1.482 1.654 | < .0001 | 0.785 | 0.746 0.825 | < .0001 |
| No (reference) | 1 | | | 1 | | |

odds people use DS. Individuals with obesity were more likely to take RXD (OR = 1.565, 95% CI 1.482–1.654) while less likely to take DS (OR = 0.785, 95% CI 0.746–0.825) compared with individuals without obesity.

Fig 2 shows the correlation between age, BMI, and the number of RDX and DS used by both the obese group and non-obese group. Fig 2(a) illustrates the correlation between the average number of RXD used by a respondent and the age groups. Generally, the average number of RXD used by a respondent increased with an increase in age for both non-obese and obese groups. Regardless of age, individuals with obesity generally take more RXD than individuals without obesity. The distribution of the average number of RXD used shows a positive skewness distribution: the average number is greater than the median within each age group. Fig 2(b) illustrates the correlation between the average number of DS used and the age groups. Similarly, the average number of DS used generally increased with an increase in age for both groups, with people aging from 65 to 74 taking the highest number of DS (mean of non-obese vs. obese: 2.47 vs.2.09). However, the difference in average number of DS used between the obese and non-obese groups was not as clear as that between the number of RXD used and age group. Fig 2(c) illustrates the correlation between the number of RXD/DS used and BMI. BMI mainly clustered around 18 to 37 $kg/m^2$. People typically take higher numbers of RXD than DS. As the BMI categories from 83 to 87 $kg/m^2$ and from 128 to 132 $kg/m^2$ have only a few persons (three persons in 83 to 87 $kg/m^2$ category and one in 128 to 132 $kg/m^2$) included in the sample, no bar shows in the figure, the points indicate the mean value among those groups.

## Classification of RDX use and DS use using machine learning

S4–S8 Tables in the Supplementary Material show the detailed results of classifying DS and RXD use. Classification of DS use (binary variable) was not as accurate as classification of RXD use (binary variable). Results from the models run were similar for DS use with the highest overall accuracy being 64.6% and the AUROC at 0.7 (S4 Table). The results for RXD use were better, with the highest overall accuracy being 74.3% and the highest AUROC at 0.816 (S5 Table). To see if results would change with the addition of the variable DS use (for RXD use classification) and RXD use (for DS classification) the same machine learning algorithms were evaluated again. The overall accuracy and AUROC increased slightly in both DS models (S4 Table) and RXD models (S5 Table). This shows that adding the extra variable did not make a significant contribution to the classifications. Using the PIR in place of the annual household income also did not significantly change the results for classification of both DS use (S6 Table) and RDX use (S7 Table). For DS classification, RXD use, insurance status, and sex were the top three important features. For RXD classification, insurance status, DS use, and age were the top three important features. We further created four groups for RDX count (i.e., 0, 1–2, 3–5, >5) and classified samples into one of these groups, using the same models and variables. In this experiment (S8 Table), the best model is logistic regression with an AUROC of 0.76. Note that multiclass classification is inherently more challenging than binary classification.

## Discussion

Individuals with obesity experience increased risk for developing chronic diseases across the lifespan, which are often managed using RXD. Overall, RXD use in the United States has increased [9], with the greatest increases seen in the treatment of conditions found to be associated with obesity, specifically antihypertensives, antihyperlipidemic, antidiabetics, and antidepressants [11]. Further, many individuals may use DS in addition to or instead of RXD.

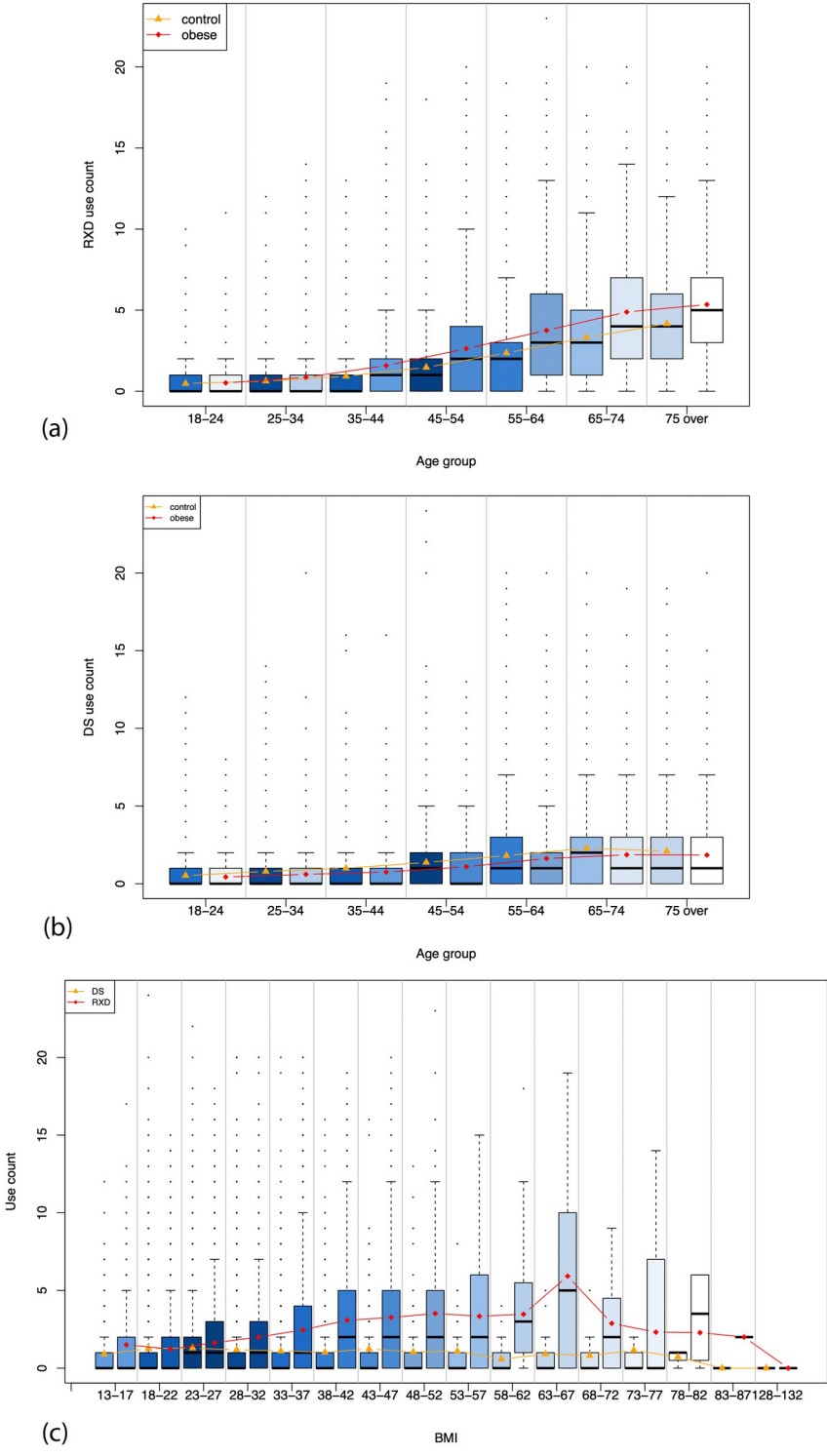

**Fig 2. (a) The correlation between the number of RXD used and age; (b) The correlation between the number of DS used and age; (c) the number of RXD/DS used compared to BMI.** The shade of color represents the aggregated weights within each age group. The points of line indicate the average number used within each age group. The shade of the color represents the aggregated weights within each age group. The points on the line indicate the average number used within each age group.

Given the increases in obesity, RXD use, and DS use, it is important to analyze trends in order to better characterize individuals RXD and DS use in relation to obesity.

In this study, we used NHANES data from 2003–2018 to examine RXD and DS use in relation to obesity status. We showed that demographics and obesity status do play a role in usage. In regard to demographic variables, we found that the obese group has a higher reported use of RXD at 63.68%. In both groups, females report higher use than males. In addition, RXD use increases with age in both groups. The difference in usage based on sex is explainable because females tend to have more consistent visits to medical practitioners and typically use more RXD in general, compared to males [39]. The increase in RDX use shown with age can be explained by increased prevalence in clinical comorbidities as the population gets older and is consistent with prior population studies [40]. The reason why the RDX usage decreases by obese individuals in the 75+ group may be that individuals whose obesity was associated with CVD earlier in life may have higher mortality rates. Another possibility is that higher BMI in older age may be protective, as previous research has suggested. Regardless, our findings further highlight that BMI is not as useful of a health parameter in older adults as it is in young and middle-aged adults, which is consistent with previous research [41]. The differences in race can be explained by the healthcare and insurance gap seen in minority races [42]. The increased use by those that are insured can be explained by an increased use of health care services and the associated increase coverage of RXD [43].

Regarding specific RXD types, cardiovascular agents and metabolic agents were used more by the obese group, while hormone/hormone modifiers and psychotherapeutic RXD usage was higher in the non-obese group. While increased use of cardiovascular and metabolic RXD was expected given the cardiometabolic comorbidities with obesity, higher usage of hormonal and psychotherapeutic RXD in non-obese individuals was surprising. Prior research has found higher rates of reproductive issues and increases in depression and other mental health disorders associated with obesity [44]; thus, we would expect higher use of related RXD. One explanation may be the association between high TSH and high BMI and low free-T4 and high BMI. These levels may not be outside of the "normal" range for these values but still cause an increase in BMI which could mean that RXD use would not necessarily be indicated [45]. In regard to psychotherapeutic RXD, it is possible that individuals with obesity may have unrecognized or undiagnosed mental health issues that are seen as medically related to obesity rather than mental health. For example, unhealthy coping mechanisms, such as increasing food intake or binging may be related to undiagnosed depressive symptomatology [46].

In looking at tracking the use of specific RXD prescribed for weight loss, there was a small proportion of the population that utilized these drugs. The problem with further study of these RXD is that many of them were approved outside of the 16-year survey data used in this project. Additionally, many of the drugs used for weight loss are also used for other purposes such as management of diabetes or as a general CNS stimulant. Based on the information available it was unclear why respondents used a specific RXD. This made it difficult to understand and analyze the use of these drugs.

When looking at both DS use and RXD use, the population with obesity was more likely to use RXD, but less likely to use DS compared with the non-obese population. This finding is consistent with a previous report that only 33.9% of adults use DS for weight loss [47]. Similar results were also obtained regardless of sex, age, and financial status [48]. However, we found additional novel predictors, such as insurance status. While higher RXD use in obese individuals was expected, lower use of DS was surprising. Given the low percentage of individuals who use DS for weight loss, it is possible that a majority of DS use is related to seeking other purported benefits. For example, there are a plethora of DS marketed toward enhancing brain

function or enhancing physical function, which may be more likely to be consumed by either older or non-obese individuals who are already health-conscious.

The experiments with machine learning models showed that predictions of RXD and DS use could be improved. Future studies should examine alternative variables that may better predict RXD and DS use. For example, diagnostic information, out-of-pocket costs, and RXD coverage information may provide valuable additional information for understanding these relationships. Validated models of RXD and DS use would provide valuable information that may inform patient education. Further, physicians and other health care providers may be able to use this information to better understand patient trends and to make informed prescribing decisions. In particular, health care providers may benefit from being able to characterize individuals most likely to use DS, as these may go unreported otherwise.

Overall, our findings indicate that obesity is associated with higher RXD and lower DS use, but income and age are important demographic factors to consider in this relationship. Since it appears that the more used RXD types are associated with comorbid conditions related to obesity instead of treating obesity directly, there may be opportunities for better health education. This may include expanding education on the benefits of lifestyle changes to minimize both obesity and the impact of common comorbid conditions. Further, the lower observed use of DS in the obese population may offer an opportunity for patient education on DS like omega-3 fatty acids and Vitamin B/B-Complex, which are shown to benefit cardiovascular health, omega-3 fatty acids which are shown to benefit weight loss and to maintain blood sugar, and multivitamin/multiminerals which are shown to impact general overall health [49]. Understanding use patterns of DS in obese and non-obese individuals may also provide opportunities for educating patients about potentially harmful or ineffective DS, such as those targeting weight loss specifically [50].

Based on Figs 1 and 2(a), there seems to be a point in the 20–30 age range that would be ideal for addressing obesity proactively and aggressively before it escalates into having comorbidities that require RXD. Additionally, there is an opportunity for better education in those under 20 as studies show that obesity in childhood and adolescence can lead to obesity as an adult [51]. Education has shifted from education on treating obesity to education on prevention of obesity, given that losing small amounts of weight or maintaining a healthy weight are more effective than treating obesity once it has developed [52]. Future studies should examine how access to healthcare may further impact these relationships, especially in regards to accessing quality obesity prevention education.

## Limitations

In most of the survey cycles used, there was no reason or diagnosis code associated with the report of RXD use. This means that we do not know why a respondent was taking a certain RXD. In addition, only primary category was used to type the various RXD. This means that if an RXD has multiple uses or off-label uses that cross-disease categories, it would not be evident in these results. Further, NHANES does not have information specifically geared towards delineating what may be considered a DS whose main purpose is weight loss. Another limitation is that NHANES is a cross-section survey. It is thus not possible to infer any casual relationships between the variables.

## Future work

Recent releases of NHANES survey cycles (2013–2018) include diagnosis codes with the RXD information. A future project that contains diagnosis codes associated with RXD use would provide more insight. Even though national health surveys like NHANES provide enormous

opportunities for answering many important health-related questions, they have not been used widely for patient education and informatics research except for a few studies of our own [53–55]. In future work, we plan to build informatics tools such as a data dashboard to visualize various types of analyses of the NHANES data to provide patients, policy makers, and health providers a way to explore RXD and DS use in the general population and certain population subgroups.

## Conclusions

As obesity becomes a larger issue and the weight crisis in the United States becomes increasingly detrimental, more needs to be done to understand the overall health status associated with this population and how to educate the public about obesity, its comorbidities, and preventive measures that may help. Knowing how RXD and DS use are different from those without obesity is only the start. Further steps can be taken to understand why there are differences and how the underlying diseases and conditions can be pre-empted in this group. Further knowledge on the association between obesity, DS, and RXD can inform patient education with the help of informatics tools such as data dashboards. Developing models that help us understand the causes of and lifestyle changes needed to change obesity status should improve overall health in the United States.

## Supporting information

**S1 Table. NHANES variables.**
(PDF)

**S2 Table. Reported prescription drug use by demographic characteristics among obese and control group.**
(PDF)

**S3 Table. Reported dietary supplements use by demographic characteristics among obese and control group.**
(PDF)

**S4 Table. Performance of machine learning models for classifying DS use.**
(PDF)

**S5 Table. Performance of machine learning models for classifying RXD use.**
(PDF)

**S6 Table. Performance of machine learning models for classifying DS use using PIR.**
(PDF)

**S7 Table. Performance of machine learning models for classifying RXD use using PIR.**
(PDF)

**S8 Table. Performance of machine learning models for classifying RXD use into categories.**
(PDF)

## Author Contributions

**Conceptualization:** Laura A. Barrett, Terrence J. Adam, Rui Zhang, Zhe He.

**Data curation:** Laura A. Barrett, Zhe He.

**Formal analysis:** Laura A. Barrett, Aiwen Xing, Julia Sheffler, Rui Zhang, Zhe He.

**Funding acquisition:** Zhe He.

**Investigation:** Laura A. Barrett, Elizabeth Steidley, Terrence J. Adam, Zhe He.

**Project administration:** Zhe He.

**Supervision:** Zhe He.

**Validation:** Aiwen Xing.

**Visualization:** Laura A. Barrett.

**Writing – original draft:** Laura A. Barrett, Elizabeth Steidley, Zhe He.

**Writing – review & editing:** Laura A. Barrett, Aiwen Xing, Julia Sheffler, Terrence J. Adam, Rui Zhang, Zhe He.

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
