## [Decision Letter · Decision Letter 0]

3 Jan 2022

PONE-D-21-29526Assessing the Use of Prescription Drugs in Obese Respondents in the National Health and Nutrition Examination SurveyPLOS ONE

Dear Dr. He,

Thank you for submitting your manuscript to PLOS ONE. After careful consideration, we feel that it has merit but does not fully meet PLOS ONE’s publication criteria as it currently stands. Therefore, we invite you to submit a revised version of the manuscript that addresses the points raised during the review process.

We look forward to receiving your revised manuscript.

Kind regards,

Jingjing Qian

Academic Editor

PLOS ONE

https://journals.plos.org/plosone/s/file?id=ba62/PLOSOne_formatting_sample_title_authors_affiliations.pdf”

“This study was supported in part by the National Institute on Aging (NIA) of the National Institutes of Health (NIH) under Award Number R21AG061431; and the University of Florida Clinical and Translational Science Institute, which is supported in part by the NIH National Center for Advancing Translational Sciences under award number UL1TR001427; and National Center for Complementary and Integrative Medicine (NCCIH) of NIH under Award Number R01AT009457. The content is solely the responsibility of the authors and does not necessarily represent the official views of the NIH.”

“This study was supported in part by the National Institute on Aging (NIA) of the National Institutes of Health (NIH) under Award Number R21AG061431 (ZH); and the University of Florida Clinical and Translational Science Institute, which is supported in part by the NIH National Center for Advancing Translational Sciences under award number UL1TR001427 (ZH); and National Center for Complementary and Integrative Medicine (NCCIH) of NIH under Award Number R01AT009457 (RZ). The content is solely the responsibility of the authors and does not necessarily represent the official views of the NIH.The funders had no role in study design, data collection and analysis, decision to publish, or preparation of the manuscript.”

Reviewers' comments:

Reviewer's Responses to Questions

**Comments to the Author**

1. Is the manuscript technically sound, and do the data support the conclusions?

Reviewer #1: Yes

Reviewer #2: Partly

2. Has the statistical analysis been performed appropriately and rigorously? 

Reviewer #1: Yes

Reviewer #2: Yes

3. Have the authors made all data underlying the findings in their manuscript fully available?

Reviewer #1: Yes

Reviewer #2: Yes

4. Is the manuscript presented in an intelligible fashion and written in standard English?

Reviewer #1: No

Reviewer #2: Yes

5. Review Comments to the Author

Reviewer #1: The authors first conducted the secondary-data analyses based on selected variables from the NHANES (2003 ~ 2014) dataset, such as: demographic, physical examination, RXD, and insurance status to figure out the association between obesity status and usage of prescription drugs. Then, machine learning models have been utilized to predict the usage of prescription drugs for people with different backgrounds.

It is a good idea to develop a machine learning model based on the findings from a national-wide survey dataset. And here are my comments:

1. Please give reference for cut-off value on BMI and age (line: 84) and type of drugs on RXDs (line: 85).

2. Please give a clear description of the variables that have been used from NHANES. For example: does RXD represent the question (RXDUSE): “ have you taken or used any prescription medicines in the past month?” And what is DS corresponding to the variables in NHANES?

3. Since the average standard for annual income across states is different, please use the Poverty Income Ratio instead of Household Income, then run the regression and machine learning model to see if the conclusion (line: 167-168) still stands.

4. Line 51: “must”. Please give a reference if the author considers obese people must take RXDs to manage these conditions. Otherwise, without any disease symptoms, lifestyle changes, such as increasing the physics activities, dietary changes are more likely to be given by the clinic Doctors.

5. Line 85: 1937/32 is the unweighted sample size for respondents, however, in Table 1, the weighted sample sizes are given. So please be consistent. I would suggest using the unweighted sample size in Table 1 to describe the basic characteristics of the study population.

6. The quality of Figure 2 needs to be improved.

7. Line 79: please consider citing more recent work, since the plural “studies” is used.

Reviewer #2: This study of using National Health and Nutrition Examination Survey (NHANES) data to assess the use of prescription drugs in obese respondents is very interesting, but there are still some concerns for this study, which are shown below.

1. What’s the purpose and importance (or impact) to predict an individual’s likelihood of using any RXD or DS, instead of specific RXD or DS? In other words, how would clinicians and patients benefit from the high or low likelihoods of using any RXD or DS, as the possibility of needing any RXD or DS instead of specific RXD or DS does not deliver clear information for both clinicians and patients. In addition, since NHANES data also provide detailed specific types for DS, why did the authors only report the specific types of RXD, but not specific types for DS?

2. The NHANES data may not be an appropriate data for prediction purpose. As the NHANES is a cross-section survey data, we do not know whether the outcome of using any RXD or DS was before or after the obesity status. In this case, if most individuals already had RXD or DS before the obesity, the prediction model constructed based on the predictors that obtained after the outcome (using RXD or DS) will not provide valid prediction performance to apply for in the real-world settings.

3. As mentioned by authors, the most recent release of NHANES survey cycle is from 2017-2018, then why did this study only use the data from 2003-2014, excluding 2015-2018? In addition, one published study by Randhawa etc. also used NHANES data from 1988-2012 to examine the medication use by body mass index and age. Compared with Randhawa’s study, what new information and scientific findings could be added and need to be discussed?

4. The methods section lacks many details. For example, how did this study identify the use of specific RXD types, DS use, BMI results, demographic and insurance information from NHANES data?

5. The authors mentioned that they performed two separate logistic regression analyses, but in the paper, only the results from the second logistic regression were shown in table 3. The results from the first logistic regression may need to be added in the supplementary materials.

6. Result section, in table one, the authors only provided the weighted data, what about the unweighted raw sample size?

7. Result section, lines 145-148, “the decrease in usage by obese individuals in the 75+ group may, in part, reflect the potential mixed effects of obesity in old age, where those with a high BMI have a lower mortality…” It is hard to understand this statement for the relationship between the lower mortality and decreased CVD drugs as well as metabolic agents. If the authors meant older patients with high BMI were less likely to develop CVD or other diseases to have better survival, then how to explain increased trend of RXD and DS use was observed in Figure 2c where high BMI (≥30) had clearly higher counts of RXD or DS?

8. Result section, lines 201-202, the largest maximum number of DS used seems among people with BMI from 23-27 kg/m2, not 18 to 22 kg/m2 based on Figure 2c.

6. PLOS authors have the option to publish the peer review history of their article (what does this mean?). If published, this will include your full peer review and any attached files.

Reviewer #1: **Yes: **You Lu

Reviewer #2: No

---

## [Author Response · Author response to Decision Letter 0]

15 Feb 2022

Response to Reviews

We would like to sincerely thank the reviewers for their constructive feedback to our manuscript. We address their comments point by point as follows:

Reviewer #1: The authors first conducted the secondary-data analyses based on selected variables from the NHANES (2003 ~ 2014) dataset, such as: demographic, physical examination, RXD, and insurance status to figure out the association between obesity status and usage of prescription drugs. Then, machine learning models have been utilized to predict the usage of prescription drugs for people with different backgrounds.

It is a good idea to develop a machine learning model based on the findings from a national-wide survey dataset. And here are my comments:

1. Please give reference for cut-off value on BMI and age (line: 84) and type of drugs on RXDs (line: 85).

Response: We added the references to the cut-off value on BMI and age. Note that we only included adult samples in this study. In this study, because the goal was to analyze the RXD use, we had to make sure that there is reported drug use information (no empty value for drug use) for these samples.

2. Please give a clear description of the variables that have been used from NHANES. For example: does RXD represent the question (RXDUSE): “ have you taken or used any prescription medicines in the past month?” And what is DS corresponding to the variables in NHANES?

Response: A new Table S1 has been added to the Supplementary Materials to give the clear description of the variables used in NHANES. 

3. Since the average standard for annual income across states is different, please use the Poverty Income Ratio instead of Household Income, then run the regression and machine learning model to see if the conclusion (line: 167-168) still stands.

Response: We extracted the PIR (poverty income ratio) from NHANES and re-ran both the regression and machine learning models using the PIR in place of the annual household income.

4. Line 51: “must”. Please give a reference if the author considers obese people must take RXDs to manage these conditions. Otherwise, without any disease symptoms, lifestyle changes, such as increasing the physics activities, dietary changes are more likely to be given by the clinic Doctors.

Response: We revised this sentence as “Many of these conditions require pharmaceutical intervention as part of the treatment plan and individuals with obesity often use RXDs to manage these conditions.”

5. Line 85: 1937/32 is the unweighted sample size for respondents, however, in Table 1, the weighted sample sizes are given. So please be consistent. I would suggest using the unweighted sample size in Table 1 to describe the basic characteristics of the study population.

Response: Table 1 has been updated to include both the weighted and unweighted sample size.

6. The quality of Figure 2 needs to be improved.

Response: High-resolution figure can be downloaded from the PDF file.

7. Line 79: please consider citing more recent work, since the plural “studies” is used.

Response: We have added more recent work using NHANES.

26. Gelfand A, Tangney CC. Analyses and interpretation of cannabis use among NHANES adults. European Journal of Preventive Cardiology. 2018 Jan 1;25(1):40–1. 

27. Yu X, Hao L, Crainiceanu C, Leroux A. Occupational determinants of physical activity at work: Evidence from wearable accelerometer in 2005-2006 NHANES. SSM Popul Health. 2022 Mar;17. 

28. Casillas A, Liang L-J, Vassar S, Brown A. Culture and cognition-the association between acculturation and self-reported memory problems among middle-aged and older latinos in the National Health and Nutrition Examination Survey (NHANES), 1999 to 2014. J Gen Intern Med. 2022 Jan;37(1):258–60. 

29. Perez-Lasierra JL, Casajus JA, González-Agüero A, Moreno-Franco B. Association of physical activity levels and prevalence of major degenerative diseases: Evidence from the national health and nutrition examination survey (NHANES) 1999-2018. Exp Gerontol. 2022 Feb;158. 

30. Wang L, Fu Z, Zheng J, Wang S, Ping Y, Gao B, et al. Exposure to perchlorate, nitrate and thiocyanate was associated with the prevalence of cardiovascular diseases. Ecotoxicol Environ Saf. 2022 Jan 6;230. 

31. Dong L, Xie Y, Zou X. Association between sleep duration and depression in US adults: A cross-sectional study. J Affect Disord. 2022 Jan 1;296:183–8. 

32. Xu Z, Gong R, Luo G, Wang M, Li D, Chen Y, et al. Association between vitamin D3 levels and insulin resistance: a large sample cross-sectional study. Sci Rep. 2022 Jan 7;12(1). 

Reviewer #2: This study of using National Health and Nutrition Examination Survey (NHANES) data to assess the use of prescription drugs in obese respondents is very interesting, but there are still some concerns for this study, which are shown below.

1. What’s the purpose and importance (or impact) to predict an individual’s likelihood of using any RXD or DS, instead of specific RXD or DS? In other words, how would clinicians and patients benefit from the high or low likelihoods of using any RXD or DS, as the possibility of needing any RXD or DS instead of specific RXD or DS does not deliver clear information for both clinicians and patients. In addition, since NHANES data also provide detailed specific types for DS, why did the authors only report the specific types of RXD, but not specific types for DS?

Response: We have changed the predictions to classifications. This can assist in informing both the patient and the practitioner about the likelihood of using RXDs based on demographics and BMI. Previous cited work covers the specifics of DS use in the population. The other reviewer commended us to use machine learning to classify DS and RXD use (“It is a good idea to develop a machine learning model based on the findings from a national-wide survey dataset.”) We revised the paper to justify the importance of classifying the likelihood of using RXD or DS as: “For example, diagnostic information, out-of-pocket costs, and RXD coverage information may provide valuable additional information for understanding these relationships. Validated models of RXD and DS use would provide valuable information that may inform patient education. Further, physicians and other health care providers may be able to use this information to better understand patient trends and to make informed prescribing decisions. In particular, health care providers may benefit from being able to characterize individuals most likely to use DS, as these may go unreported otherwise.”

2. The NHANES data may not be an appropriate data for prediction purpose. As the NHANES is a cross-section survey data, we do not know whether the outcome of using any RXD or DS was before or after the obesity status. In this case, if most individuals already had RXD or DS before the obesity, the prediction model constructed based on the predictors that obtained after the outcome (using RXD or DS) will not provide valid prediction performance to apply for in the real-world settings.

Response: We have changed the predictions to classifications. Hence, we are not trying to predict RXD use but classify where someone is compared to others in the same group. We do not go into the causality of taking specific medications in this paper or how long someone has taken a specific RXD. We do mention that the use of specific RXD may be the result of being obese or may be the cause of increased weight gain leading to obesity. 

3. As mentioned by authors, the most recent release of NHANES survey cycle is from 2017-2018, then why did this study only use the data from 2003-2014, excluding 2015-2018? In addition, one published study by Randhawa etc. also used NHANES data from 1988-2012 to examine the medication use by body mass index and age. Compared with Randhawa’s study, what new information and scientific findings could be added and need to be discussed?

Response: Randhawa et al. looked at changes in medication use over time (1988-2012) to see if obesity or age were factors. They focused significantly on the different drug categories. They focused more on the causality of the increase in RXD usage from 1988 to 2012. We used the data collectively instead of based on collection cycle. We examined multiple factors beyond just age and obesity status. While we touch on the different categories, overall, that was not the focus of the study. We did not seek to address the cause of the increase in RXD use, we were more interested in the comparison of the control group versus the non-control group over the different variables and the connection to the use of DS.

4. The methods section lacks many details. For example, how did this study identify the use of specific RXD types, DS use, BMI results, demographic and insurance information from NHANES data?

Response: A table indicating what fields from NHANES were used has been added to the Supplementary Materials Table S1.

5. The authors mentioned that they performed two separate logistic regression analyses, but in the paper, only the results from the second logistic regression were shown in table 3. The results from the first logistic regression may need to be added in the supplementary materials.

Response: The result of the first regression can be found in Table S2 and Table S3 in the supplementary material. The result of the second regression can be found in Table 3 in the main text.

6. Result section, in table one, the authors only provided the weighted data, what about the unweighted raw sample size?

Response: Table 1 has been updated to include both the weighted and unweighted amounts.

7. Result section, lines 145-148, “the decrease in usage by obese individuals in the 75+ group may, in part, reflect the potential mixed effects of obesity in old age, where those with a high BMI have a lower mortality…” It is hard to understand this statement for the relationship between the lower mortality and decreased CVD drugs as well as metabolic agents. If the authors meant older patients with high BMI were less likely to develop CVD or other diseases to have better survival, then how to explain increased trend of RXD and DS use was observed in Figure 2c where high BMI (≥30) had clearly higher counts of RXD or DS?

Response: This section has been revised as “(In Results Section Page 10) The decrease in usage by obese individuals in the 75+ group may, in part, reflect the potential mixed effects of obesity in old age. Studies indicate that higher than normal BMI can have a lower mortality in the 75+ age group, though this is dependent on many other variables.37–39 ….... (In Discussion Section Page 18) The reason why the RDX usage decreases by obese individuals in the 75+ group may be that individuals whose obesity was associated with CVD earlier in life may have higher mortality rates. Another possibility is that higher BMI in older age may be protective, as previous research has suggested. Regardless, our findings further highlight that BMI is not as useful of a health parameter in older adults as it is in young and middle-aged adults, which is consistent with previous research.42”

8. Result section, lines 201-202, the largest maximum number of DS used seems among people with BMI from 23-27 kg/m2, not 18 to 22 kg/m2 based on Figure 2c.

Response: According to the raw data, the largest maximum number of DS used is among people with BMI from 18-22 kg/m2.

---

## [Decision Letter · Decision Letter 1]

16 Mar 2022

PONE-D-21-29526R1Assessing the Use of Prescription Drugs in Obese Respondents in the National Health and Nutrition Examination SurveyPLOS ONE

Dear Dr. He,

Thank you for submitting your manuscript to PLOS ONE. After careful consideration, we feel that it has merit but does not fully meet PLOS ONE’s publication criteria as it currently stands. Therefore, we invite you to submit a revised version of the manuscript that addresses the points raised during the review process.

We look forward to receiving your revised manuscript.

Kind regards,

Jingjing Qian

Academic Editor

PLOS ONE

Journal Requirements:

Additional Editor Comments:

Dear authors, we appreciate your effort in addressing reviewers' comments. However, please directly respond to reviewer #2's comment regarding not adding newer NHANES 2015-2018 data in your manuscript. The current analysis using the 2003-2014 NHANES data is dated and might compromise implications of your findings.

In addition, I suggest you to further revise the title and/or abstract to reflect the objective of the paper. Your objective was to examine the use of prescription drugs and dietary supplements by the individuals with obesity, but the current title and abstract (results section) only focus on use of prescription drugs. The aspect of dietary supplements is missing in title and abstract (results section). Please also double check the results in the abstract to reflect the most up to date data based on the revision. Thanks.

Reviewers' comments:

Reviewer's Responses to Questions

**Comments to the Author**

1. If the authors have adequately addressed your comments raised in a previous round of review and you feel that this manuscript is now acceptable for publication, you may indicate that here to bypass the “Comments to the Author” section, enter your conflict of interest statement in the “Confidential to Editor” section, and submit your "Accept" recommendation.

Reviewer #1: All comments have been addressed

Reviewer #2: (No Response)

2. Is the manuscript technically sound, and do the data support the conclusions?

Reviewer #1: Yes

Reviewer #2: (No Response)

3. Has the statistical analysis been performed appropriately and rigorously? 

Reviewer #1: Yes

Reviewer #2: (No Response)

4. Have the authors made all data underlying the findings in their manuscript fully available?

Reviewer #1: Yes

Reviewer #2: Yes

5. Is the manuscript presented in an intelligible fashion and written in standard English?

Reviewer #1: Yes

Reviewer #2: (No Response)

6. Review Comments to the Author

Reviewer #1: (No Response)

Reviewer #2: The authors addressed almost all comments, except for comment 3 which was not fully addressed. The authors may need to update the analysis including the most recent data cycles (2015-2018) which are available now. In addition, the Figure 2C is hard to tell the statement "the largest maximum number of DS used is among people with BMI from 18-22 kg/m2", so this figure may need some modification and improvement.

7. PLOS authors have the option to publish the peer review history of their article (what does this mean?). If published, this will include your full peer review and any attached files.

Reviewer #1: No

Reviewer #2: No

---

## [Author Response · Author response to Decision Letter 1]

25 Apr 2022

Thanks for your feedback! We have addressed the comments point-by-point as follows.

Journal Requirements:

***** Response: We have reviewed the reference list to make sure it is complete and correct. 

Additional Editor Comments:

Dear authors, we appreciate your effort in addressing reviewers' comments. However, please directly respond to reviewer #2's comment regarding not adding newer NHANES 2015-2018 data in your manuscript. The current analysis using the 2003-2014 NHANES data is dated and might compromise implications of your findings.

***** Response: We have added the data of NHANES survey cycles 2015-2016, 2017-2018 in the revised version. We have re-done all the analyses and updated the results accordingly. 

In addition, I suggest you to further revise the title and/or abstract to reflect the objective of the paper. Your objective was to examine the use of prescription drugs and dietary supplements by the individuals with obesity, but the current title and abstract (results section) only focus on use of prescription drugs. The aspect of dietary supplements is missing in title and abstract (results section). Please also double check the results in the abstract to reflect the most up to date data based on the revision. Thanks.

***** Response: We revised the title and abstract to reflect the objective of the paper. 

Reviewer #2: The authors addressed almost all comments, except for comment 3 which was not fully addressed. The authors may need to update the analysis including the most recent data cycles (2015-2018) which are available now. In addition, the Figure 2C is hard to tell the statement "the largest maximum number of DS used is among people with BMI from 18-22 kg/m2", so this figure may need some modification and improvement.

***** Response: We have added the data of NHANES survey cycles 2015-2016, 2017-2018 in the revised version. We have re-done all the analyses and updated the results accordingly. For better clarity, we removed the two sentences “The largest maximum number of DS used (24) can be found among people with BMI from 18 to 22 kg/m2. The highest average number of RXD used can be found among people with BMI from 63 to 67 kg/m2.”

---

## [Decision Letter · Decision Letter 2]

18 May 2022

Assessing the Use of Prescription Drugs and Dietary Supplements in Obese Respondents in the National Health and Nutrition Examination Survey

PONE-D-21-29526R2

Dear Dr. He,

We’re pleased to inform you that your manuscript has been judged scientifically suitable for publication and will be formally accepted for publication once it meets all outstanding technical requirements.

Kind regards,

Jingjing Qian

Academic Editor

PLOS ONE

Additional Editor Comments (optional):

Thank you for further revising your manuscript by incorporating new years of data.

Reviewers' comments:

Reviewer's Responses to Questions

**Comments to the Author**

1. If the authors have adequately addressed your comments raised in a previous round of review and you feel that this manuscript is now acceptable for publication, you may indicate that here to bypass the “Comments to the Author” section, enter your conflict of interest statement in the “Confidential to Editor” section, and submit your "Accept" recommendation.

Reviewer #2: All comments have been addressed

2. Is the manuscript technically sound, and do the data support the conclusions?

Reviewer #2: Yes

3. Has the statistical analysis been performed appropriately and rigorously? 

Reviewer #2: Yes

4. Have the authors made all data underlying the findings in their manuscript fully available?

Reviewer #2: Yes

5. Is the manuscript presented in an intelligible fashion and written in standard English?

Reviewer #2: (No Response)

6. Review Comments to the Author

Reviewer #2: The authors may need to double check the submitted Table S2 in their supporting information. The numbers are not consistent with the text in the manuscript 'RXD ( = 0.53, 95% CI 0.498-0.564, = 0.569, 95% CI 0.52-0.623).' It seems the authors still used the old table S2, as the numbers are same as the numbers deleted in the track-changing version.

7. PLOS authors have the option to publish the peer review history of their article (what does this mean?). If published, this will include your full peer review and any attached files.

Reviewer #2: No

---

## [Editor Report · Acceptance letter]

23 May 2022

PONE-D-21-29526R2 

Assessing the Use of Prescription Drugs and Dietary Supplements in Obese Respondents in the National Health and Nutrition Examination Survey 

Dear Dr. He:

I'm pleased to inform you that your manuscript has been deemed suitable for publication in PLOS ONE. Congratulations! Your manuscript is now with our production department. 

Kind regards, 

on behalf of

Dr. Jingjing Qian 

Academic Editor

PLOS ONE